# MOTE-NAS: Multi-Objective Training-based Estimate for Efficient Neural Architecture Search

**Yu-Ming Zhang**[1]    **Jun-Wei Hsieh**[2]    **Xin Li**[3]    **Ming-Ching Chang**[3]
**Chun-Chieh Lee**[1]    **Kuo-Chin Fan**[1]
[1]National Central University    [3]University at Albany
[2]National Yang Ming Chiao Tung University
108522036@g.ncu.edu.tw; jwhsieh@nycu.edu.tw

## Abstract

Neural Architecture Search (NAS) methods seek effective optimization toward performance metrics regarding model accuracy and generalization while facing challenges regarding search costs and GPU resources. Recent Neural Tangent Kernel (NTK) NAS methods achieve remarkable search efficiency based on a training-free model estimate. However, they overlook the non-convex nature of the DNNs in the search process. In this paper, we develop **Multi-Objective Training-based Estimate (MOTE) for efficient NAS**, retaining search effectiveness and achieving the new state-of-the-art in the accuracy and cost trade-off. To improve NTK and inspired by the Training Speed Estimation (TSE) method, MOTE is designed to model the actual performance of DNNs from macro to micro perspective by drawing the loss landscape and convergence speed simultaneously. Using two *reduction strategies*, the MOTE is generated based on a reduced architecture and a reduced dataset. Inspired by evolutionary search, our iterative ranking-based, coarse-to-fine architecture search is highly effective. Experiments on NASBench-201 show MOTE-NAS achieves 94.32% accuracy on CIFAR-10, 72.81% on CIFAR-100, and 46.38% on ImageNet-16-120, outperforming NTK-based NAS approaches. An evaluation-free (EF) version of MOTE-NAS delivers high efficiency in only **5 minutes**, delivering a model more accurate than KNAS.

## 1   Introduction

Neural Architecture Search (NAS) [52] tackles the challenge of automating the design and search for suitable neural network architectures in many applications. NAS approaches mainly comprise two stages: a model *search stage* dedicated to identifying promising candidates within the architecture search space, and an *evaluation stage* where candidate performance is assessed. In the search stage, the search space can be exponentially large. To reduce search complexity, the cell-based tabular search space [49, 10, 38] is widely considered. Reinforcement learning [52, 2, 39] and evolutionary algorithms [27, 31, 33, 46, 9] are also used to accelerate the search process. However, the primary cost of NAS lies in the evaluation stage, where candidate models must undergo intensive training until convergence to obtain a precise performance assessment. This incurs significant time costs (*e.g.*, NASBench-201 requires 3-10K GPU seconds for convergence after 200 epochs). So, various *proxy estimates* (e.g., zero-cost proxy [1] and training speed estimation [34]) have been developed to rank candidates, mitigating computational demands for model evaluation.

Recently, several estimates based on NTK have been proposed, including TE-NAS [6], KNAS [47], and Eigen-NAS [51]. NTK-based estimates serve as condensed representations of gradients and their correlations. The NTK theory aims to macro-model the actual performance of Deep Neural Networks (DNNs). It assumes that the performance of an infinite-width DNN can be fully described by the

38th Conference on Neural Information Processing Systems (NeurIPS 2024).

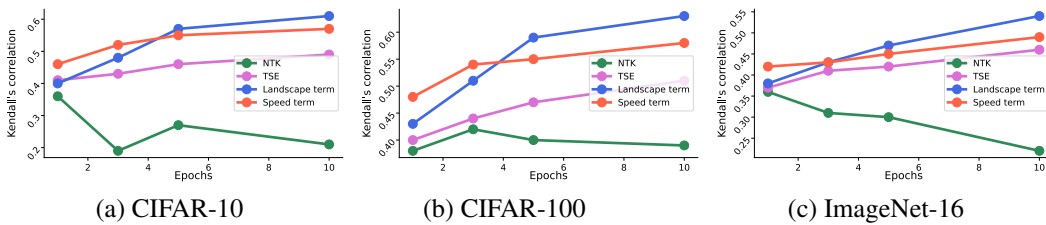

| (a) CIFAR-10 | (b) CIFAR-100 | (c) ImageNet-16 |

Figure 1: Post-training rank correlation for randomly chosen 1000 candidates on NASBench-201. The predictive performance of the proposed two terms gradually improves as epochs increase.

NTK at initialization and that the NTK's value remains unchanged after training [19]. Consequently, NTK-based estimates can predict the actual performance of candidate architectures without training. However, real DNNs have limited width and exhibit a highly non-convex nature, leading the NTK to encounter significant nonlinear changes during training and resulting in limitations in accurately predicting the actual performance of candidates. Fig. 1 shows the NTK suffers unstable performance during training.

To address this macro-modeling issue, we propose a novel *landscape term* that leverages the idea from the study [12] to linearly combine the differences between the initial weights and the post-training weights of the candidate architectures, which allows us to capture the non-convex nature of these candidates by the landscape slice. If the landscape is flatter, the candidate's performance tends to be better, as it more readily converges to the global optima. Furthermore, studies from a micro-aspect have been conducted to model this issue, such as Snip [20], Grasp [41], and SynFlow [40, 1], which use gradient change to predict the performance of candidate architectures. As the gradient is integrated into the model training, it may reflect the current effectiveness of changes in the model training. Similarly, TSE [34] directly sums up the training loss to represent the convergence speed to predict candidate performance. In summary, these methods are more intuitive. Although they may not theoretically capture the macroscopic non-convex nature of DNNs, in practice, as shown in Fig. 1, the performance TSE even exceeds NTK. This observation inspired us to propose a *speed term* that sums the training loss per unit of time, providing a microscopic description of the convergence speed of candidates.

This paper introduces a Multi-Objective Training-based Estimate (MOTE) that considers *landscape term* for macroscopic view and *speed term* for microscopic view to predict the performance of candidates in a joint optimization. This dual perspective offers a comprehensive consideration and an accurate estimate for candidates. Furthermore, we introduce two reduction strategies to reduce the time costs by generating MOTE, which consists of landscape and speed terms. We also present MOTE by integrating the evolutionary algorithm [32], named MOTE-NAS. This design is based on a coarse-to-fine iterative procedure for architecture search. In the search stage of MOTE-NAS, we maintain several dozen to several hundred promising candidate architectures in the pool. In the evaluation stage, we first select the top-$K$ architectures based on the MOTE, then select the best architecture by the early stopping version of the test accuracy. We further develop a stripped-down, evaluation-free version named MOTE-NAS-EF, which achieves high efficiency and can finish a NAS run in merely eight minutes.

MOTE-NAS outperforms mainstream NTK-based NAS methods. Fig. 2 compares it with TE-NAS [6], KNAS [47], Eigen-NAS [51], and Label-Gradient Alignment (LGA) [30] on CIFAR-100 of NASBench-201. The accuracy of the final architecture discovered by MOTE-NAS is significantly superior to other methods. In our experiments, MOTE-NAS achieved 94.32% accuracy on CIFAR-10, 72.81% on CIFAR-100, and 46.38% on ImageNet-16-120. The evaluation-free version, MOTE-NAS-EF, achieves results comparable to KNAS's, where the search is completed in only eight minutes. The technical contributions of this work are summarized as follows:

- Our proposed NAS approach utilizes an efficient training-based estimate to optimize landscape view and convergence speed objectives jointly. This design comprehensively captures the non-convex nature of DNNs from a macro perspective and monitors the convergence speed from a micro perspective, enabling precise actual performance predictions for desired architectures.

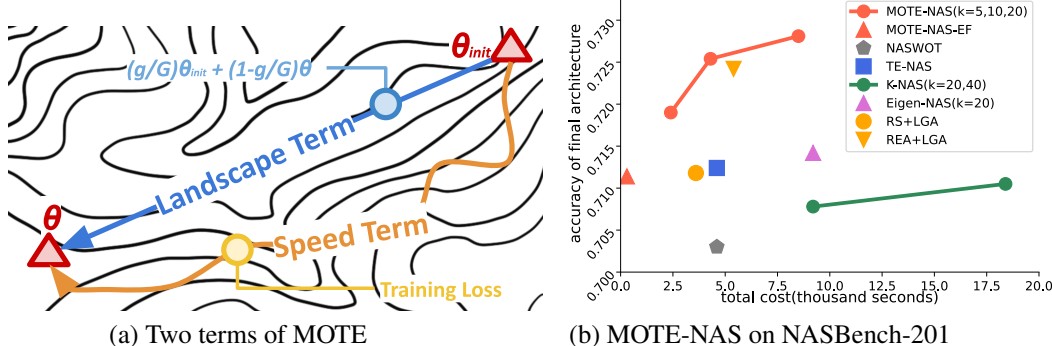

| | |
|---|---|
| (a) Two terms of MOTE | (b) MOTE-NAS on NASBench-201 |

Figure 2: (a) The **landscape term** draws the slice of loss landscape to capture its macroscopic non-convex nature of the candidate architecture. The **speed term** analyzes the training changes over the training time, providing microscopic insights into the candidate's convergence speed. (b) Comparison of **MOTE-NAS** and an **Evaluation-Free version MOTE-NAS-EF** against other recent efficient NAS methods on NASBench-201 (CIFAR-100).

- To enable lightweight training-based estimates, we introduce two reduction strategies for speeding up MOTE generation. Unlike other benchmarks, a readuced meta-architecture is used, and the proposed reduced dataset is built by selecting representative labels of CIFAR-100.
- Our MOTE-NAS achieves a new state-of-the-art for NAS in the accuracy-cost plot (refer to Fig. 2(b)). A stripped-down, evaluation-free version of MOTE-NAS is highly efficient, with performance of the resulting model still outperforming some NTK-based methods, such as the KNAS.

## 2 Related Work

**DARTS.** Instead of exploring a discrete set of architectures, the *Differentiable ARchiTecture Search (DARTS)* [26] transforms the combinatorial challenge of finding optimal operations into a continuous optimization problem within a differentiable search space. A notable challenge with DARTS is the potential dominance of easily optimized operators, such as skip-connections and pooling operations in the early stages. This issue impedes selecting more potent operations like convolutions with large kernels. In [7, 22], a robust prior is introduced to restrict the number of skip connections within a cell to a predefined value. The progressive search strategy employed in P-DARTS [7] gradually increases network depth and refines candidate operations based on mixed operation weight. DARTS methods are efficient when running with limited computational resources. However, the architecture found comes with stability and generalizability issues. Furthermore, DARTS algorithms often prefer shallow and wide structures [37].

On the other hand, NAS methods generally comprise two stages: the search stage and the evaluation stage. The former focuses on collecting promising candidate architectures, while the latter involves assessing the performance of these candidate architectures.

**Search Stage.** Numerous studies have concentrated on the search stage. Some NAS methods leverage reinforcement learning [52, 2, 39], while others are rooted in evolutionary algorithms [27, 31, 33, 46, 9]. The cell-based tabular search space [49, 10, 38] is effective in reducing exhaustic search into a more manageable scale, using a meta-architecture with predefined operations, hyperparameters, filters, and strides. The candidate architectures under consideration range from tens to hundreds of thousands of candidate architectures. Recently, **predictor-based** approaches [24, 28, 43, 11, 42, 45, 15] have gained popularity. These methods construct predictors trained with architecture-accuracy pairs to forecast the performance of a candidate architecture. These predictors encompass a range of models from graph convolutional networks [17] to MLPs and other regression models. However, obtaining a high-quality set of architecture-accuracy pairs for NAS is non-trivial.

**Evaluation Stage.** Compared with the cost of the search stage, the burden of NAS mainly resides in the time-consuming evaluation process. Various studies have proposed proxy estimates to reduce the need for a real performance evaluation. A prominent approach is zero/few-cost estimate [1, 34,

23, 29, 6, 47, 51, 30], which substitutes performance indicators such as accuracy with alternative estimates. The zero-cost proxy [1] introduces zero-cost performance estimates [20, 41, 40] and TSE [34] introduces a training speed estimate. More recently, Neural Tangent Kernel (NTK)-based estimates [6, 47, 51, 30] have gained popularity based on the assumption that DNNs can predict their convergence performance at initialization. However, it is experimentally found in [30] that NTKs cannot capture the non-linear characteristics of DNN training dynamic well. Recent NAS methods integrate multiple approaches to achieve remarkable performance. For instance, OMNI [44] and ProxyBO [36] propose few-cost NAS methods by combining zero-cost estimates with more resource-intensive techniques like Bayesian optimization and performance predictors.

## 3 MOTE-NAS

### 3.1 Multi-Objective Training-based Estimate

NTK theory tries to describe gradient change by a macro-perspective, but its fundamental assumption about infinite-width DNN cannot fit the real DNNs that have finite width. For example, in KNAS [47], GKH asserts the existence of a gradient feature that can serve as a coarse-grained proxy to support downstream training when evaluating randomly initialized architectures. However, this does not propose a concrete solution to identify such a non-linear gradient feature during training. LGA[30] finds that the sensitivity for weight initialization that leads to NTK cannot perform stably, demonstrating that the value of NTK does not change [19]. In addition, Fig. 1 shows that NTK do not accurately predict the actual performance of candidates in practice. Despite NTK-based estimates, there are estimates that make predictions by micro-perspective. Among them, TSE[34] sums up the training loss as a proxy estimate to represent the convergence speed. The convergence speed as an important factor for model performance has been extensively discussed in the literature [13, 16, 34]. Specifically, these studies inspired the proposal of MOTE. MOTE introduces a new *landscape term* to capture the non-convex nature of models by a macro-perspective through the linear combination between two weights to observe the loss landscape. Simultaneously, MOTE introduces another new *speed term* to measure the convergence speed of the model by a micro-perspective through calculating the unit time training loss. Incorporating the multi-objectives enables MOTE to comprehensively describe candidates' non-convex nature and convergence speed from a macro-to-micro perspective.

**Landscape Term.** In order to capture non-convexity of loss landscape by macro-perspective, we introduce *landscape term* that linearly combines the two weights before and after few-training to interpolate the weights for intermediate state, and then calculates the loss values (cross-entropy) of these weights, which means to cutting a section from loss landscape to observe its nature [12], so we sum these loss values to determine whether the loss landscape is smooth, detailed follows.

Let $\theta$ denote the trained weights of the candidate. To macro-model the actual performance of candidate architectures, we linearly combine the initial weights with the trained weights $\theta$ to obtain the combined weights, denoted $\theta(g)$, for describing the non-convex landscape of the candidates. Then,we obtain the combined weights $\theta(g)$ in the form:

$$\theta(g) = (\frac{g}{G})\theta_{init} + (1 - \frac{g}{G})\theta, \tag{1}$$

where $\theta_{init}$ denotes the initial weights, and $G$ is the number represent how dense the linear combination and set to be 10 based on most of the experiments from [12]. Let $Y_{pd}^{\theta(g)}$ denote the model prediction labels based on the weights $\theta(g)$, and $Y_{gt}$ be the ground-truth labels of the training data. Then, we use cross-entropy to measure the difference between $Y_{pd}^{\theta(g)}$ and $Y_{gt}$. After that, we sum the loss value of these middle weights as *landscape term* as follows.

$$\sum_{g=0}^{G} \mathcal{J}_{\theta(g)} = \sum_{g=0}^{G} CE(Y_{pd}^{\theta(g)}, Y_{gt}). \tag{2}$$

The poposed landscape term can capture the non-convexity of models where a lower value indicates a flatter loss landscape, implying an efficient convergence of the candidate and avoiding the problem of sharp minimum [50].

**Speed Term.** On the other hand, the idea to model the actual performance of candidates by micro-perspective, such as TSE, still perform strong and important, as shown in Fig. 1. Therefore, we are

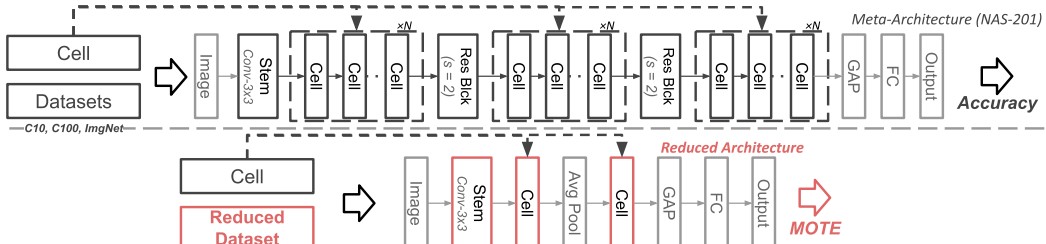

Figure 3: The generation pipelines of accuracy (upper part) and MOTE (bottom part). The proposed reduced architecture and dataset, MOTE, are colored red in their respective sections.

inspired by TSE [34] to introduce *speed term*. It first calculates training losses (cross-entropy) over an epoch, and divides it by the time cost of an epoch, converting it into a unit time training loss to measure convergence speed, which means that it observes the convergence speed of the candidates under standardized time expenditure, which helps to standardize the measurement of architectures of different sizes. It can also be seen from the Fig. 1 that the speed term performs better than the TSE [34]. The detailed *speed term* is defined:

$$\sum_{e=1}^{E} \frac{l_e}{t_e} = \sum_{e=1}^{E} \frac{CE(Y_{pd}^e, Y_{gt})}{t_e},\qquad(3)$$

where $Y_{pd}^e$ denotes the model prediction labels in epoch $e$, and $Y_{gt}$ means the ground truth labels of the training data. Then, we use cross-entropy to calculate the loss between $Y_{pd}^e$ and $Y_{gt}$. After that, we divide the loss value by the time cost $t_e$ within epoch $e$ and then sum up all as *speed term*. Note that a lower value indicates faster convergence, which could imply better performance.

Now, MOTE integrates the *landsacpe term* and *speed term* to model actual performance of candidates from macro to micro perspective, and is defined as follows:

$$MOTE = f(\sum_{g=0}^{G} \mathcal{J}_{\theta(g)}) + f(\sum_{e=1}^{E} \frac{l_e}{t_e}),\qquad(4)$$

where the first term is the proposed *landscape term*, the latter term is the proposed *speed term*, where $\mathcal{J}_\theta$ denotes the function used to determine whether the loss landscape is smooth by linear combining initial weights $\theta_{init}$ and trained weights $\theta$. $E$ is the number of maximum training epochs to search candidate architectures, $l_e$ is the training loss(usually measured by cross-entropy) for the $e$th epoch, $t_e$ denotes the time cost for the epoch $e$, and the function $f$ denotes the non-linear transformation to restrict the range of values.

Due to the different ranges of *landscape term* and *speed term*, we use the box-cox transformation [3] to transform and normalize them; more comparisons of other transformation methods are detailed in **Appendix A.1**. MOTE can consider both objectives by adding transformed method to assess their combined impact. Since lower values for both *landscape term* and *speed term* suggest a potentially better performance of the models, a smaller MOTE value indicates a better performance in practical application. Fig. 2(a) illustrates the concepts of *landscape term* and *speed term*.

## 3.2 Reduction Strategies for MOTE Generation

MOTE requires little training to obtain *landscape term* and *speed term*, which makes it crucial to balance minimizing training time and ensuring adequate training change. Consequently, we introduce a more compact meta-architecture called the reduced architecture. We also propose the reduced dataset method, which involves a smaller dataset built by CIFAR-100. MOTE combines these reduction strategies and the training-based objectives introduced earlier to produce promising estimates with few costs. Fig. 3 depicts this process.

**Reduced Architecture (RA).** MOTE is not the actual performance of DNN after combining the cell with meta-architectures. Instead, MOTE is generated from the change in weight and loss acquired during training. The generation of MOTE relies on the proposed reduced architecture. The reduced architecture is designed to eliminate redundant layers from most meta-architectures[49, 10], resulting

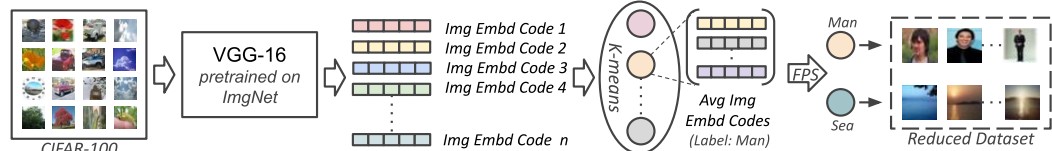

Figure 4: After encoding the images of CIFAR-100 through VGG, the encodings for each label are obtained by averaging image embedding codes. Then we used K-Means and Farthest Point Sampling (FPS) to select a representative set of $r$ labels, forming the reduced dataset.

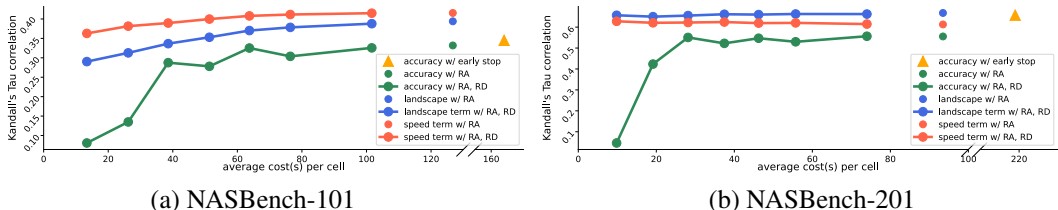

(a) NASBench-101            (b) NASBench-201

Figure 5: The proposed terms via aggressive reduction strategies on NASBench-101 and NASBench-201. RA means reduced architecture, RD means reduced dataset

in a simplified network structure to save the time cost of each epoch. It only comprises a convolutional layer as the stem layer and two cell layers and employs aggressive downsampling through a pooling layer with a kernel size of $4 \times 4$. This architectural simplification significantly accelerates the training process and substantially reduces the cost of obtaining MOTE. The structure of the reduced architecture is shown at the bottom of Fig. 3.

**Reduced Dataset (RD).** To minimize the computational cost of MOTE generation, we propose a sub-dataset derived from CIFAR-100 [18], referred to as the reduced dataset. As shown in Fig. 4, the process of constructing the reduced dataset involves several steps: 1) We use a VGG-16 model pre-trained on ImageNet-1K [8] to extract logits from images; 2) Flattened the logits and averaged them according to specific label, resulting in label embedding codes; 3) K-Means to cluster the label embedding codes into $r$ groups; 4) Farthest Point Sampling (FPS) to select $r$ label embedding codes from $r$ group to represent the $r$ labels of the reduced dataset; 5) The images associated with the chosen $r$ labels are reserved for building the reduced dataset. The reduced dataset is a proxy dataset and a sub-dataset of CIFAR-100, containing a representative set of $r$ labels. When $r$ is set to 100, the reduced dataset is equivalent to CIFAR-100. As $r$ decreases, the reduced dataset becomes smaller and easier to fit for candidate models. However, regardless of the value of $r$, the reduced dataset maintains the original image distribution for each label. The K-means and FPS techniques ensure that the reduced dataset represents a significant part of CIFAR-100 even when $r < 100$. The most important thing is MOTE generation that rely on a reduced dataset can save remarkable time cost.

**Two Terms of MOTE with Reduction Stratigies.** To further observe how reduction strategies work, we randomly selected 1K candidates of NASBench-201 as toy experimental subjects. As shown in Fig. 5, the early stop version of test accuracy (after 12 epochs) has a high correlation with test accuracy (after 200 epochs), but training then getting it required about 220 gpu seconds, which is still a remarkable cost. When RA is applied, the time cost decreases 60%, but the correlation also drops to about 0.5 from 0.65. After further RD is applied, as the extraction ratio $r$ gradually decreases (the further to the left the smaller $r$ is), the time cost is also greatly saved, but the correlation suffers intolerable losses. The correlation of the leftmost ($r = 10$) is even less than 0.1. In contrast, the proposed *landscape term* and *speed term* always maintain a high correlation when applying RA and RD. As the extraction ratio $r$ gradually decreases, the time cost is reduced to about 10 gpu seconds from 220 seconds, and the correlation is still about 0.65. This is because the two proposed terms based on weight and loss changes do not rely on the excluded middle state of true or false. More comparisons between various reduction strategies refer to **Appendix A.2**.

### 3.3 Integrating MOTE with Evolutionary Search

Although MOTE consistently maintains impressive performance under the influence of the reduction strategy, MOTE remains a proxy estimate. A comprehensive NAS method still requires the participation of accuracy to evaluate the actual performance of the candidates discovered in the evaluation

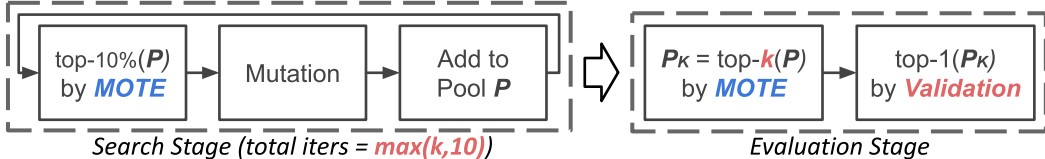

Search Stage (total iters = *max(k,10)*)          Evaluation Stage

Figure 6: The left side depicts MOTE-NAS's search stage, utilizing MOTE for architecture selection through an evolutionary loop, terminating at $10 + k$ iterations. On the right side is the evaluation stage, where MOTE selects the top-$k$ architectures for evaluation. MOTE-NAS-EF simplifies this by relying solely on MOTE to choose the top-1 architecture without the evaluation stage.

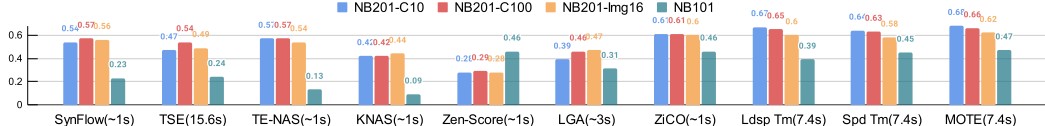

Figure 7: The Kendall's Tau Correlation comparison of the proposed speed term, landscape term and MOTE with other estimates on NASBench-101 and NASBench-201. Note that the "(s)" is the GPU seconds per cell cost.

stage, similar to previous NAS methods [6, 47, 51, 30]. However, as mentioned at the outset, the time cost of accuracy is exceedingly high, often demanding thousands of GPU seconds. Therefore, our proposed MOTE-NAS first employs MOTE to assist the evolutionary process in obtaining a small subset of promising candidates. Then, through the early stopping version of accuracy assessment, the best is identified. The entire procedure is illustrated in Fig. 6.

In the search stage, MOTE-NAS maintains a pool of promising candidates $P$, with batch size $B$ initially set to 10. With the continuous evolution loop, every ten cycles, $B$ is incremented by 10. In each evolutionary process, MOTE is generated to sort $P$ and take the top 10% of the candidates to the mutation stage. For the mutation stage, we are inspired by predictor-based methods [43, 11, 45] to encode candidates into adjacency and operation matrices. Subsequently, we calculate the Euclidean distance between each pair of candidates to select up to $10 \times k$ mutated candidates, which are then added to $P$. This evolutionary loop continues $max(k, 10)$ times, then stops and enters the evaluation stage. For the evaluation stage, MOTE is used to select the top $k$ architectures of $P$ ($k = 5, 10, 20$), then select their best architecture based on the early topping version of the test accuracy.

## 4 Experimental Results

**Experimental Setup.** We used NASBench-101 and NASBench-201, both cell-based search spaces. NASBench-101 has 423,621 candidates trained on CIFAR-10 for 108 epochs. NASBench-201 includes 15,625 candidates trained on CIFAR-10, CIFAR-100, and ImageNet-16-120 for 200 epochs each. Computation was on Tesla V100 GPUs, with MOTE or MOTE-NAS costs calculated specifically on V100. Our experiment had three parts: comparing MOTE with other estimates on NASBench-101 and NASBench-201, evaluating MOTE-NAS against other NAS methods on NASBench-201, and using MOTE-NAS to search for a mobilenet-like architecture on ImageNet-1K. Further, we visualize the rankings of MOTE and KNAS to perceive their differences in Fig. 8. MOTE is generated from the proposed reduced architecture and dataset. We used reduced dataset with a sampling rate hyperparameter $r = 10$ based on the results in Fig. 5. The hyperparameters are batch size 256, epochs 50, learning rate 0.001 with Adam optimizer, and cross-entropy loss function. MOTE generation per cell took about seven GPU seconds under these settings.

### 4.1 Comparison of MOTE and Other Estimates

To explore the performance gaps between MOTE and other relevant estimates, we compared NASBench-101 and NASBench-201. We ranked candidates using speed term, landscape term and MOTE or other estimates and compared the resulting rankings to the actual ranking, calculating Kendall's Tau correlation to gauge the performance of these estimates. The experimental results are presented in Fig. 7.

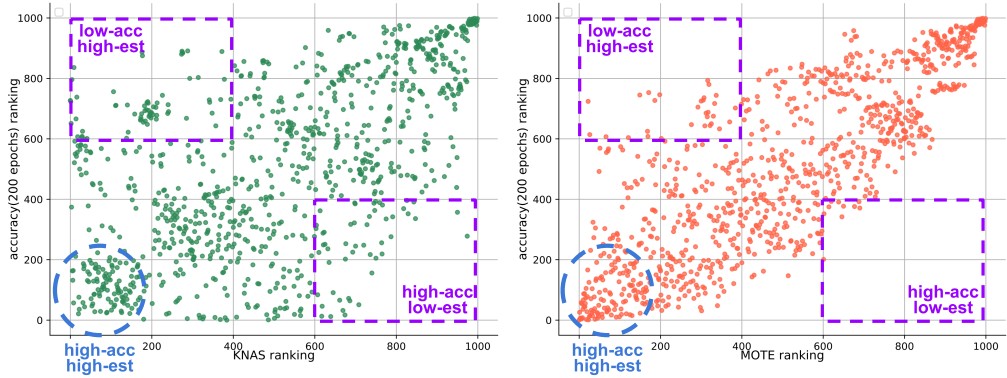

Figure 8: Comparison of the distribution of MOTE (red) and KNAS (green) on NASBench-201 (CIFAR-100).

**NTK-based Estimates.** MOTE leverages the landscape term to capture the non-convex nature from the candidate architectures, compensating for the shortcomings in NTK-based estimates. As shown in Fig. 7, MOTE achieves correlations of 0.68, 0.66, 0.62, and 0.47 in NASBench-101 and NASBench-201. Compared to TE-NAS and KNAS, MOTE shows performance improvements ranging from 13% to 62% on NASBench-201. In addition, we observed poor performance of TE-NAS and KNAS in NASBench-101, with KNAS demonstrating a correlation of merely 0.09. This illustrates that NTKs struggle to adapt to larger search spaces and more diverse candidate architectures in environments like NASBench-101. In contrast, MOTE maintains a high correlation of 0.47, demonstrating a significant increase of 261% and 422% compared to TE-NAS and KNAS, respectively. Compared to the state-of-the-art ZICO, MOTE still performs better than it does on benchmarks. Remarkably, these MOTE performance gains were achieved with an average cost of only seven seconds per candidate.

**Other Estimates.** When comparing MOTE with other estimates such as SynFlow [40, 1] and TSE [34], MOTE maintains a significant advantage. Compared to SynFlow and TSE, MOTE shows performance gains of 11% to 45% on NASBench-201 and 96% and 104% on NASBench-101, respectively. It should be noted that both TSE and MOTE are training-based estimates, and MOTE accelerates $2\times$ faster than TSE, outperforming it significantly.Additionally, we can see that the proposed speed term and landscape term also perform well in Fig. 7, both of which are essential components of MOTE.

## 4.2 Visualization of MOTE and NTK based Estimate

We depicted the distribution of MOTE in comparison to KNAS to facilitate analysis regarding MOTE and NTK-based estimates. As shown in Fig. 8, the experimental results involve random selection of 1K candidate architectures from NASBench-201 (CIFAR-100). The x-axis represents the estimate ranking based on MOTE or KNAS, while the y-axis represents the actual ranking based on the test accuracy after 200 epochs. Each node in the figure represents a candidate architecture, with its (x, y) coordinates indicating its position in the estimate and the actual rankings, respectively. Both the estimate ranking and the actual ranking are sorted from high to low scores.

In the left subfigure of Fig. 8, we present the distribution of KNAS, while the right subfigure displays the distribution of MOTE. In general, MOTE is more concentrated than KNAS, indicating that MOTE is closer to the actual ranking of the candidates than KNAS. This observation aligns with the superior performance of MOTE in Kendall's Tau correlation comparisons, as shown in Fig. 7. Further, focusing on the high-performance region (the blue circle in the lower left corner of the two sub-figures), MOTE exhibits a more concentrated trend compared to the chaotic distribution of KNAS. This suggests that MOTE outperforms KNAS in predicting promising architectures. Finally, examining the purple boxes in the lower right and upper left corners of the two subfigures reveals candidates for which the estimate indicates good. However, the actual performance is poor, or vice versa. In both cases, MOTE's misjudgments are significantly fewer than those of KNAS, visually confirming that MOTE is highly competitive compared to the mainstream NTK-based estimate.

Table 1: Comparison of the proposed MOTE-NAS and others on NASBench-201. Note that 'Cost (s)' indicates the cost in seconds calculated on Tesla V100. Entries in bold with underlines indicate the best performance, and those in bold alone represent the second-best performance.

| Type | Model | CIFAR-10 | | CIFAR-100 | | ImgNet-16 | |
|------|-------|----------|--|-----------|--|-----------|--|
| | | Acc(%) | Cost(s) | Acc(%) | Cost(s) | Acc(%) | Cost(s) |
| Predictor | Neural Predictor [43] | 94.07 | 840K | 72.18 | 840K | 46.39 | 2.4M |
| | Arch-Graph [15] | - | - | 73.38 | 840K | - | - |
| | WeakNAS [45] | 94.23 | 840K | 73.42 | 840K | 46.79 | 2.4M |
| | Proxy-BO [36] | - | - | **73.48** | 1.2M | **47.18** | 3.2M |
| Few-Cost | NASWOT [29] | 92.96 | 2.2K | 70.03 | 4.6K | 44.43 | 10K |
| | TE-NAS [6] | 93.90 | 2.2K | 71.24 | 4.6K | 42.38 | 10K |
| | KNAS (k=20) [47] | 93.38 | 4.4K | 70.78 | 9.2K | 44.63 | 20K |
| | KNAS (k=40) [47] | 93.43 | 8.8K | 71.05 | 18.4K | 45.05 | 40K |
| | Eigen-NAS (k=20) [51] | 93.46 | 4.4K | 71.42 | 9.2K | 45.53 | 20K |
| | RS + LGA [30] | 94.05 | 5.4K | 71.56 | 7.0K | 46.30 | 15K |
| | REA + LGA [30] | **94.30** | 3.6K | 72.42 | 5.4K | 45.30 | **3.6K** |
| | **MOTE-NAS (k=5)** | 93.97 | **2.2K** | 71.89 | **2.4K** | 46.10 | 5.8K |
| | **MOTE-NAS (k=10)** | 94.15 | 4.2K | **72.54** | 4.3K | **46.38** | 11.3K |
| | **MOTE-NAS (k=20)** | **94.32** | 8.5K | **72.81** | 8.5K | **46.34** | 22.7K |
| | **MOTE-NAS-EF** | 93.54 | **0.5K** | 71.59 | **0.6K** | 44.73 | **0.6K** |

## 4.3 Comparisons of MOTE-NAS and Other NAS

To compare the performance differences between MOTE-NAS and other NAS methods, we carried out experiments on NASBench-201. Tab. 1 presents the experimental results, where "Acc(%)" represents the accuracy of the final architecture discovered by the NAS methods on the test set of the respective dataset. At the same time, "Cost(s)" indicates the total seconds used by NAS methods to discover this final architecture.

**MOTE-NAS with Top-$k$ Evaluation.** MOTE-NAS combines MOTE with an evolutionary algorithm to filter and mutate potential high-scoring candidates by MOTE. Ultimately, the top $k$ high-scoring candidates are selected using the early stopping version of the test accuracy (after 12 epochs), similar to the approaches in [47, 51]. The time consumption of MOTE-NAS lies in training candidates to obtain MOTE during the search stage and the early stopping accuracy obtained during the evaluation stage. However, the cost of each MOTE is only about seven gpu seconds, so primary consumption is still to evaluate the top-$k$ candidates. We have set the $k$ range from $5, 10, 20$. When $k = 5$, the final architecture found by MOTE-NAS achieves significantly higher accuracy on three datasets of NASBench-201, compared to TE-NAS[6], KNAS[47], and Eigen-NAS[51], with speedups ranging from $1.9\times$ to $6.9\times$. It only slightly lags behind REA + LGA[30] in test accuracy. However, when $k = 10$ or $k = 20$, with a time consumption of 8.5K gpu seconds, the final architecture discovered by MOTE-NAS achieves a remarkable accuracy of 94.32% on CIFAR-10 and 72.81% on CIFAR-100. Moreover, the MOTE-NAS discovered candidate architecture achieves 46.38% on ImageNet-16 with 11.3K seconds. Compared to NTK-based NAS (TE-NAS, KNAS, Eigen-NAS, LGA), the proposed MOTE-NAS consistently achieves the best accuracy with the lowest cost.

**Assessing the Evaluation-Free Version of MOTE-NAS.** The proposed MOTE-NAS has shown impressive performance in balancing time consumption and efficiency. The substantial time cost led us to consider omitting the evaluation stage to pursue a faster MOTE-NAS framework. Especially considering that MOTE, compared to other estimates, achieves a higher Kendall's Tau correlation, indicating a significant improvement in MOTE's predictive performance. Hence, omitting additional validation information became a viable option. To accomplish this, we removed the entire evaluation stage from MOTE-NAS. At the end of the search stage, we utilized MOTE to select top-1 as the final architecture. This variant is referred to as MOTE-NAS-EF in Tab. 1.

Although MOTE-NAS-EF experiences an accuracy loss, the search cost savings are notable. MOTE-NAS-EF achieved 93.54% accuracy on CIFAR-10, 71.59% on CIFAR-100, and 44.73% on ImageNet-16-120 with the search cost of only about 0.6K gpu seconds. In contrast, KNAS requires 4.4K, 18.4K, and 20K seconds to achieve accuracies of 93.38%, 71.05%, and 44.63%, respectively. In particular, MOTE-NAS-EF matches KNAS in accuracy but accelerates the process by $4.8\times$ to $22.2\times$, underscoring the superiority of MOTE-NAS-EF in speed.

Table 2: This table shows the top-1 accuracy of architectures found on ImageNet using MOTE-NAS and other NAS methods.

| Model | MFLOPs | Top-1(%) | Cost(d) |
|-------|--------|----------|---------|
| MobileNetV2 [35] | 300 | 71.5* | N/A |
| MobileNetV3 [14] | **220** | 74.1* | - |
| OFA [4] | 406 | **77.7** | 50 |
| BN-NAS [5] | 470 | 75.7 | 0.8 |
| NASNet-B [53] | 488 | 72.8 | 1800 |
| CARS-D [48] | 496 | 73.3 | 0.4 |
| ZICO [21] | 448 | 75.8* | 0.4 |
| **MOTE-NAS** | **387** | **76.2** | **0.1** |
| **MOTE-NAS** | 473 | **77.1** | **0.1** |

## 4.4 MOTE-NAS on ImageNet-1K

**Search Space.** We search for a promising architecture based on the mobilenetV3 search space using MOTE, then train and evaluate it on imagenet-1K. The mobilenetV3 search space is a open search space that has five inverted residual blocks with the SE module. Every block has several hyperparameters, such as the expansion ratio for input channel expansion, kernel size, and SE module attached or not. Based on it, we restrict the selection range for each hyperparameter. We restrict the expansion ratio range from $2, 4, 6$, kernel size range from $3, 5, 7$, and the SE module used or not.

**Rescaled Reduced Architecture for Macro-Search.** Our study introduces a Rescaled Reduced Architecture for Macro-Search, where we modify the reduced architecture to accommodate the simultaneous assessment of five blocks and their collective performance. By expanding the cell layers from two to five and independently sampling the structure of each layer, our rescaled approach enables macro-search capabilities beyond single-cell exploration. Further technical specifics of this rescaled reduced architecture are outlined in Appendix A.5. Subsequently, employing MOTE-NAS with this modified architecture, we conducted a search within the mobilenetv3 space under approximately 400M FLOPs. Following 200 epochs of training using 10 GTX 2080Ti GPUs on the imagenet-1K dataset, the results (see Table 2) demonstrate the efficacy of our approach. While the accuracy of MOTE-NAS (76.2% and 77.1%) trails slightly behind OFA's 77.7%, MOTE-NAS achieves this with a significantly reduced computational cost of 0.1 GPU days compared to OFA's 50 days, representing a 500x speed improvement. Furthermore, our retraining of ZICO's best architecture yielded a 75.8% accuracy on imagenet, surpassed by MOTE-NAS with its superior accuracies and a 4x faster search cost than the 0.4 day of ZICO.

## 5 Conclusion

In this paper, we design a novel training-based estimate for efficient Neural Architecture Search (NAS) from a multi-objective optimization perspective. The key idea is to use landscape terms to capture the non-convex nature of candidate architectures from a macro perspective, and use speed terms to monitor convergence speed from a micro perspective into the estimated design. The proposed MOTE efficiently generates the landscape and speed terms with two reduction strategies, which wisely trade-off the consideration of architecture and dataset. These designs can effectively capture the non-linear characteristics of deep neural network training, address the drawbacks of NTK methods, and achieve a new state-of-the-art state. We extend our approach by iterative ranking-based evolutionary search, then deduce an evaluation-free version (MOTE-NAS-EF) that runs even faster. Extensive experimental results demonstrate the superiority of our new NAS methods over other frontier NAS methods, including KNAS, LGA, and ZICO, on NASBench-101, NASBench-201, and ImageNet-1K. Future works include expanding MOTE to other NAS frameworks, such as predictor-based methods, to pursue precise search results while exploring MOTE's generalization ability. Another line of extension is to work with the more challenging NASBench-301 benchmark dataset [38] that offers a much larger and more complex architecture space than both NASBench-101 and NASBench-201.

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

# A  Appendix

Table A1: **Kendall's $\tau$ correlation** between the MOTE scoring output and the test accuracy(after 200 epochs). Experiments are performed to compare four versions of MOTE on three sub-datasets (CIFAR-10, CIFAR-100, ImgNet-16) of the NASBench-201.

|  | CIFAR-10 | CIFAR-100 | ImgNet-16 |
|---|---|---|---|
| MOTE w/o transformation | 0.66 | 0.64 | 0.59 |
| MOTE w/ logarithm | 0.66 | 0.64 | 0.60 |
| MOTE w/ reciprocal | 0.67 | 0.65 | 0.60 |
| **MOTE w/ box-cox** | **0.68** | **0.66** | **0.62** |

## A.1  Non-linear Transformation for MOTE

The proposed Multi-Objective Training-based Estimate (MOTE) achieves efficient Network Architecture Search (NAS) based on two essential components, namely the *landscape* and *speed* terms. These terms and their variants play a pivotal role in modeling the training dynamics and thus serve as a performance estimate for candidate architectures during the search. To address the considerable variability of these terms, in the main paper, we propose to incorporate the box-cox method in the MOTE design to constrain both terms. We next provide additional experimental results to support such a design by evaluating the influence of other designs and variants of these terms. Specifically, we compute kandall's correlation on NASBench-201 to compare four versions of MOTE that contain logarithm, reciprocal, box-cox, and the original.

Table A1 shows the comparative results of four versions of MOTE. MOTE w/ box-cox performs the best. The box-cox transformation leads to a notable improvement compared to the original in correlation: 2% improvement for CIFAR-10, 2% improvement for CIFAR-100, and 3% improvement for ImageNet-16-120. These improvements are attributed to the ability to contract outliers, bringing them closer to the normal data distribution and reducing the impact on MOTE.

## A.2  Analysis of Two Reduction Strategies

Although the proposed MOTE-NAS approach can greatly reduce the required NAS cost in producing a suitable network architecture, the time cost of model training and evaluation is still the bottleneck and trade-off. We further analyze the two proposed reduction strategies and present the results of the ablation study regarding the computation time trade-off. Fig. 3 provides the MOTE generation pipeline that contains the reduced architecture and the reduced dataset.

**Experiments setting.** We conducted a random sampling of 1,000 candidate architectures from NASBench-101 and NASBench-201, respectively. Notably, these candidate architectures consist of their predefined meta-architecture and cells within their search spaces. Each cell shares identical hyperparameters and operations. Therefore, to obtain complete models, the candidate architectures (equivalent to candidate cells) must be assembled into either the meta-architecture or our reduced architecture. The proposed reduced architecture speeds up model search by retaining only a single convolutional layer as the stem layer and two cell layers, eliminating the majority of layers from the original meta-architecture of NASBench-101 and NASBench-201. The reduced dataset is a representative proxy subset of CIFAR-100, with the aim of accelerating the training process by reducing the number of training images, thus reducing the time needed for each training epoch.

**Reduced Architecture vs. Meta-Architecture.** Table A2 compares the model parameters and FLOPs of predefined meta-architecture and the peoposed reduced architecture, when evaluated on the NASBench-101 and NASBench-201 benchmarks. For the NASBench-101 case, the reduced architecture contains much fewer parameters of 34.4K, in contrast to the 2.3M parameters of the original meta-architecture. This is a significant reduction of 64×. Similarly, the FLOPs for the reduced architecture amount to 18.3M, while the original meta-architecture records 1.0G FLOPs, indicating an acceleration of 435×. In the case of NASBench-201 candidates, the reduced architecture exhibits reductions/accelerations of 15× and 4× compared to the original meta-architecture. These results indicate a superior advantage of the proposed reduction strategy in both memory requirements and computational speed on the two benchmarks.

Table A2: **Comparisons of memory and computation requirements** between our reduced architecture and original predefined meta-architecture of NASBench-101 and NASbench-201.

|  | Params | FLOPs |
|---|---|---|
| Meta-Architecture (NAS101) | 2.3M | 1.0G |
| **Reduced Architecture (NAS101)** | **34.4K** | **18.3M** |
| Meta-Architecture (NAS201) | 388.7K | 52.7M |
| **Reduced Architecture (NAS201)** | **25.7K** | **13.6M** |

Table A3: **Ablation study for three reduced strategies on NASBench-101.** $r$ represents the number of selected labels in the reduced dataset. Cost(s) indicates the average seconds for each candidate to obtain the estimate. $\tau(\%)$ is the Kendall's $\tau$ correlation.

|  | $r$ | T-FLOPs | Cost(s) | $\tau(\%)$ |
|---|---|---|---|---|
| Acc w/ MA | N/A | 190.7B | 163.1 | 34.5 |
| Acc + RA | N/A | 8.7B | 127.1 | 8.1 |
| MOTE + RA | N/A | 8.7B | 127.1 | 50.1 |
| MOTE + RA+ RD | 80 | 7.0B | 103.0 | 49.5 |
| MOTE + RA+ RD | 60 | 5.2B | 78.2 | 48.8 |
| MOTE + RA+ RD | 40 | 3.5B | 56.8 | 47.8 |
| MOTE + RA+ RD | 20 | 1.7B | 26.3 | 46.8 |
| MOTE + RA+ RD | 10 | 872.6G | 13.2 | 46.6 |

**Ablation Study of Reduction Strategies.** We outline the experimental setup for two reduction strategies: (1) a reduced architecture inspired by predefined meta-architecture of NASBench-101 and NASBench-201, (2) a reduced dataset for efficiently training. Evaluations are performed on the NASBench-101 benchmark. We rank 1,000 randomly selected candidate architectures under different reduction conditions. The $\tau$ correlation measures MOTE and test accuracy obtained from model training after 108 epochs. The early stopping test accuracy after 4 epochs serves as the baseline. We incrementally introduce the reduced architecture and reduced dataset to assess the performance variations induced by the two strategies. Let $T$ denote the number of training images, $e$ denote the total epochs in search, and $f$ denote the FLOPs of the candidate architecture. The total FLOPs (T-FLOPs) required throughout the training process are calculated as:

$$\text{T-FLOPs} = T \times e \times f. \tag{A1}$$

Table A3 presents the results of the ablation study. In the table, "Acc w/ MA" denotes the early stopping test accuracy (after 4 epochs) from training on the original meta-architecture. When utilized it for candidate ranking, it achieves a correlation of 34.5%. However, owing to the complexity nature of the meta-architecture, it incurs substantial time costs, with T-FLOPs reaching 190.7B and a time cost of 88.2 seconds per candidate. In contrast, "Acc w/ RA" denotes the replacement of the meta-architecture with a reduced architecture, leading to a considerable reduction in time cost to 8.7B T-FLOPs and 29.2 seconds. However, this time reduction comes with a significant performance loss, as the correlation drops to 8.1%, primarily attributed to the sensitivity nature of the test accuracy.

Finally, "MOTE + RA" represents our proposed solution that addresses trade-off concerns. Using MOTE's dual objectives, it maintains the same time cost while elevating the correlation to 50.1%, even surpassing the baseline. Subsequently, "MOTE + RA + RD" indicates the further introduction of the reduced dataset, which reduces the number of training images used in the search. The time cost rapidly decreases with the smaller values of the hyperparameter $r$. At $r = 10$, it requires only 872.6G T-FLOPs and 13.2 seconds, resulting in acceleration rates of $21.7\times$ and $27.5\times$ compared to the baseline, with a slight decrease in correlation to 46.6%.

## A.3 Implementation Details of MOTE-NAS

Despite MOTE consistently delivering impressive performance aided by reduction strategies, it remains only a proxy estimate. A comprehensive NAS method still requires the participation of

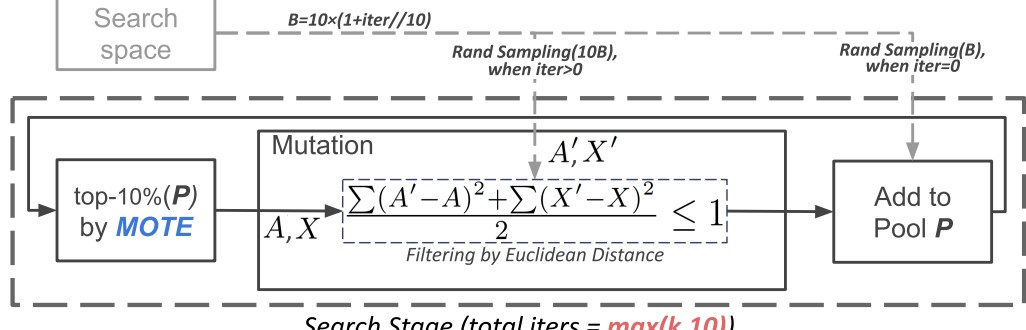

Figure A1: **The mutation step of the evolutionary process in the search.** When the iteration count is 0, as opposed to being greater than 0, the number of samples taken from the search space varies. The sampling size $B$, increases with the number of iterations.

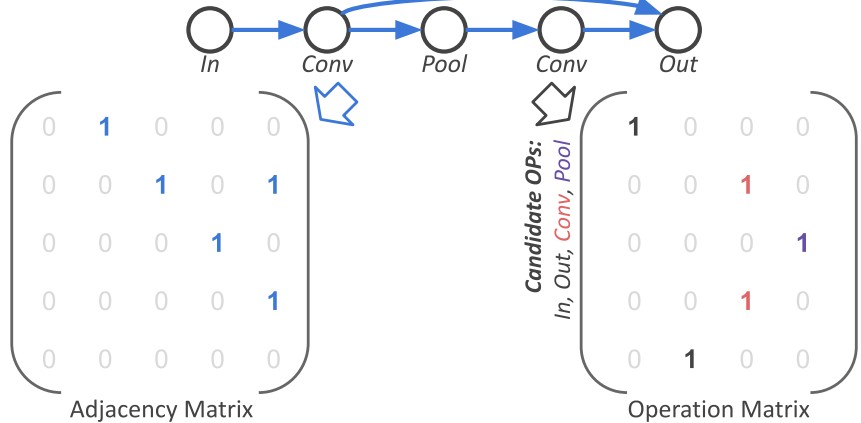

Figure A2: **Encoding a cell into the *adjacency* and *operation* matrix.** This entire procedure closely resembles the predictor-based NAS.

accuracy to evaluate the actual performance of the candidates discovered, similar to NTK-based NAS methods [6, 47, 51]. However, the time cost to obtain test accuracy is exceedingly high, often requiring thousands of GPU seconds. Hence, we employ MOTE to assist the evolutionary process in the search stage to obtain a small subset of potentially promising candidates, and then evaluate them by test accuracy. As illustrated in Fig. A1, throughout the iterative evolutionary process, MOTE-NAS relies on mutations to quickly transform known promising architectures (in pool $P$) into candidates for the next iteration. The detailed steps of the mutation process are outlined as follows:

1. Select the top 10% candidate architectures from the pool $P$ based on MOTE.
2. Encode candidate architectures $S$ into corresponding adjacency matrices $A$ and operation matrices $X$.
3. Mutate $S$ to obtain the mutated architectures $S'$. The encoded matrices $A'$ and $X'$ derived from $S'$ satisfy $\frac{\sum(A'-A)^2+\sum(X'-X)^2}{2} \leq 1$.
4. If the size of $S'$ exceeds $10 \times k$, randomly select $10 \times k$ and add them to the pool $P$. Otherwise, add $S'$ to $P$.

Fig. A2 illustrates the methodology employed in Step 2 to encode candidate architectures. In the cell-based search space [25, 49, 10, 38], candidate architectures are decomposed into smaller cells with predefined hyperparameters, including the total number of nodes within a cell, the maximum number of edges, and the allowable operations for each node. The fixed nature of these hyperparameters ensures the consistent shape of each cell. This consistency allows to represent edge connections within the cell using an adjacency matrix, as well as using one-hot vectors to represent the selected operations of each node using an operation matrix. Generating these fixed-shape matrices for all

Table A4: **Comparison of the proposed MOTE-NAS and others on NASBench-201.** Cost (s) indicates the cost in seconds calculated on Tesla V100. Acc (%) represents the accuracy produced by the resulting network on respective datasets. Entries in bold with underlines indicate the best performance, and those in bold alone represent the second-best performance.

| Model | CIFAR-10 | | CIFAR-100 | | ImgNet-16 | |
|---|---|---|---|---|---|---|
| | Acc(%) | Cost(s) | Acc(%) | Cost(s) | Acc(%) | Cost(s) |
| NASWOT [29] | 92.96 | **2.2K** | 70.03 | 4.6K | 44.43 | 10K |
| TE-NAS [6] | 93.90 | **2.2K** | 71.24 | 4.6K | 42.38 | 10K |
| KNAS (k=20) [47] | 93.38 | 4.4K | 70.78 | 9.2K | 44.63 | 20K |
| KNAS (k=40) [47] | 93.43 | 8.8K | 71.05 | 18.4K | 45.05 | 40K |
| Eigen-NAS (k=20) [51] | 93.46 | 4.4K | 71.42 | 9.2K | 45.53 | 20K |
| RS + LGA [30] | 94.05 | 5.4K | 71.56 | 7.0K | __46.30__ | 15K |
| REA + LGA [30] | __94.30__ | 3.6K | **72.42** | 5.4K | 45.30 | __3.6K__ |
| **MOTE-NAS-RS (k=5)** | 93.71 | **1.5K** | 71.59 | **1.7K** | 44.95 | **5.1K** |
| **MOTE-NAS-RS (k=10)** | 93.93 | 3.0K | 72.11 | **3.1K** | 45.84 | 10.1K |
| **MOTE-NAS-RS (k=20)** | **94.07** | 6.1K | __72.60__ | 6.8K | **46.13** | 20.7K |

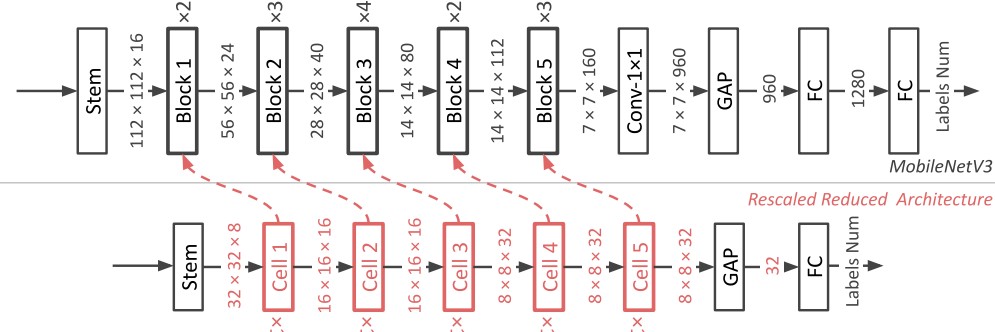

Figure A3: The bottom one is the proposed rescaled reduced architecture, it consists of five independent cells that have different structure than other four.

candidates enables the representation of structural differences by calculating element-wise distances between corresponding matrices of two candidates and summing the results.

## A.4 MOTE-NAS with Random Sampling

We remove the evolutionary search stage in MOTE-NAS and apply Random Sampling (RS) to test the performance of MOTE under various sampling strategies. This hybrid approach, termed MOTE-NAS-RS, initially draws $100 \times k$ candidates from the search space by random sampling. The top-$k$ candidates are then selected based on MOTE, further refined to top-1 based on early stopping test accuracy (after 12 epochs). Our experiments on NASBench-201 involved 10 runs for each experiment, with the results averaged and presented in Table A4.

Although MOTE-NAS-RS naturally lags behind MOTE-NAS due to the inherent simplicity of random sampling compared to the evolutionary algorithm, MOTE-NAS-RS notably outperforms mainstream NTK-based NAS. Achieving a final architecture accuracy of 94.07%, MOTE-NAS-RS surpasses KNAS [47], Eigen-NAS [51], and RS+LGA [30] on CIFAR-10. On CIFAR-100, it reaches an accuracy of 72.60%, even outpacing REA+LGA [30] at 72.42%. For ImgNet-16-120, it closely trails RS+LGA. This highlights the effectiveness of MOTE, demonstrating superior performance even when paired with a basic sampling strategy. Future studies may explore the potential combination of MOTE with NTK-based NAS without training, as suggested by [30].

## A.5 Rescaled Reduced Architecture for MobileNetV3 Search Space

To analyze the performance of MOTE-NAS in the open search space, we used it to search for promising architectures in the mobilenetv3 search space. Mobilenetv3 search space contains five inverted residual blocks with the SE module, on the other word, the search targets are five different

cells, but the proposed reduced architecture is only for single cell search. Therefore, we rescale the reduced architecture to achieve this new purpose, as shown in Fig. A3, the rescaled version consists of five different cells to fit five blocks while still maintaining a lightweight scale. It allows us to search an approximate open search space, and the architecture MOTE-NAS found has shown remarkable performance on imageNet-1K, the results refer to Tab. 2.

