# OpenReview forum: "MOTE-NAS: Multi-Objective Training-based Estimate for Efficient Neural Architecture Search"
_NeurIPS.cc/2024/Conference — NeurIPS 2024 poster_

### Official Review · Reviewer_33pQ · 2024-06-25

**Soundness:** 4
**Presentation:** 4
**Contribution:** 3
**Rating:** 6
**Confidence:** 4

**Summary:**

This paper proposes Multi-Objective Training-based Estimate (MOTE) for efficient NAS, leveraging landscape view and convergence speed to estimate the performance of neural architectures. It also introduces two reduction strategies for speeding up MOTE generation. Compared to other training-free NAS methods, MOTE achieves better search performance on NasBench201.

**Strengths:**

+ The paper is well-written.
+ The search cost of NAS is very low, enabling NAS with very limited computing resources.
+ The theoretical analysis is sound.
+ The experimental results are promising.

**Weaknesses:**

- Code is not available.
- The experimental results on ImageNet are not very good.
- Regarding the title "Multi-Objective Training-based Estimate for Efficient NAS," on NasBench201, only accuracy is considered as a metric, missing other important metrics like the number of parameters, FLOPs, latency, etc.
- The reduction strategies lack novelty, as many low-fidelity estimation methods have been proposed, including early stopping[1], training with down-scaled models[1], and training on a subset of the data[2].



[1]: B. Zoph, V. Vasudevan, J. Shlens, and Q. Le. Learning transferable architectures for scalable image recognition. In Conference on Computer Vision and Pattern Recognition (CVPR), 2018.

[2]: A. Klein, S. Falkner, S. Bartels, P. Hennig, and F. Hutter. Fast bayesian optimization of machine learning hyperparameters on large datasets. In Artificial intelligence and statistics, pages 528–536. PMLR, 2017.

**Questions:**

I would like to ask about the results on ImageNet in Table 2. OFA costs 50 days, including the training and search process. Do you consider the training cost of the searched architectures? I think 0.1 days is far from enough to well-train a model on ImageNet.

**Limitations:**

No additional significant limitations. Please refer to the weaknesses. It would be beneficial if the paper could add future research directions for this work and extend this method beyond image classification tasks.

---

> ### Author Rebuttal · Authors · 2024-08-05
>
> Thank you very much for your valuable feedback and positive evaluation of our work.
>
> **Answers to Questions:**
>
> **A1.** Yes, by adjusting the K value in MOTE-NAS to extend the search time, we can achieve better architectures. While the current architecture obtained through our method does not have the highest accuracy on ImageNet, the significantly lower time cost is a considerable advantage.
>
> **Answers to Weaknesses:**
>
> **AW1.** We have included the core code of MOTE in the supplementary materials, and the complete code will soon be available on GitHub (the link is not shown due to the policy of NIPS 2024).
>
> **AW2.** Please refer to A1.
>
> **AW3.** Generally, current NAS research focuses on finding the architecture with the highest accuracy within a given search space (NASBench-101 and NASBench-201) as quickly as possible. Other factors such as parameters, FLOPs, latency, and other metrics can be considered in our cost function in a hybrid form.
>
> **AW4.** The reduction strategies serve as acceleration methods for MOTE. When MOTE applies RD+RA, it does not experience significant performance loss in terms of "accuracy." MOTE can achieve substantial acceleration with minimal performance loss and still outperform other NAS methods, as demonstrated in Fig. 7 and Tab. 1 of our paper. Therefore, we emphasize the robustness of MOTE rather than the novelty of the reduction strategies. In TABLE III of the attached file, we replaced the random selection-based RD-RS, and MOTE still performed well. This further demonstrates that the novelty of the reduction strategies is not a primary factor affecting MOTE's performance.

---

> > ### Comment · Reviewer_33pQ · 2024-08-12
> > **Comments Added After Reading Author Response**
> >
> > Thank you for the further response. In your response to question A1, you mentioned that the training cost of the searched architecture has been considered. However, I am concerned that 0.1 GPU days may not be sufficient for training neural architectures to convergence on ImageNet. Could you please clarify if the 0.1 GPU days include the training time? Thank you.

---

> > > ### Author Response · Authors · 2024-08-12
> > >
> > > Thank you for your reply. The experiments conducted on ImageNet only represent the search time and do not include the training time. Since the MOTE-NAS-EF (top-k=1) version we used in Tab.2 is evaluation-free, we specifically emphasized that this method requires only 0.1 GPU days for the search process. In fact, training this top one architecture found after the search still takes nearly a week on a single V100 GPU.

---

> > > > ### Comment · Reviewer_33pQ · 2024-08-12
> > > >
> > > > Thank you for your response. If the searched architecture requires additional training, it may be unfair to compare the search time directly with the OFA in Table 2, as it includes both search and training costs.

---

> > > > > ### Author Response · Authors · 2024-08-12
> > > > >
> > > > > Thank you for your feedback. NAS methods can generally be categorized into the following two types:
> > > > >
> > > > > 1. **Zero-shot**: These methods rely entirely on performance estimates for ranking and only select the top-1 candidate as final architecture. In other words, the search process for these NAS methods does not require any "training". This is why some methods (CARS-D, ZiCO, and Ours) in Tab. 2 exhibit such impressive search speeds.
> > > > >
> > > > > 2. **Multi-shot**: These methods do not solely depend on performance estimates for ranking. Instead, they involve training (or evaluating) several promising candidates during the search process and then selecting the one with the best actual performance as the final architecture. In other words, these methods require multiple or iterative "training" of several candidates to achieve a more stable final architecture, which explains why their cost appears much higher (OFA).
> > > > >
> > > > > Therefore, the costs reported in Tab. 2 reflect the search time of each method, providing a fair comparison. The difference in costs arises because some NAS methods incorporate "training" during the search process while others do not.

---

> > > > > > ### Comment · Reviewer_33pQ · 2024-08-12
> > > > > >
> > > > > > Thanks for your answers. According to the Table.1 in OFA paper [1], the search cost is 40 hours,  50 days in table 2 is not consistent with the [1].
> > > > > >
> > > > > >
> > > > > >
> > > > > >
> > > > > >
> > > > > >
> > > > > > [1]Han Cai, Chuang Gan, Tianzhe Wang, Zhekai Zhang, and Song Han. Once-for-all: Train one network and specialize it for efficient deployment. arXiv preprint arXiv:1908.09791, 2019.

---

> ### Author Response · Authors · 2024-08-12
>
> Thank you for your feedback. The discrepancy arises due to differences in the search space and other hyperparameter settings. The configuration in our Tab. 2 was based on the experimental setup from ZiCO [1] (refer to their Table 2). Some of the values marked with an asterisk in our Tab. 2 are experimental results that we have reproduced.
>
> [1] Li, Guihong, et al. "Zico: Zero-shot nas via inverse coefficient of variation on gradients." arXiv preprint arXiv:2301.11300 (2023).

---

> > ### Comment · Reviewer_33pQ · 2024-08-12
> >
> > Thank you again for your response. Although the search cost of 50 GPU days for OFA in ZiCO does not make sense to me, your answer has partially addressed my concerns. I will keep a positive score for this work.

---

> > > ### Author Response · Authors · 2024-08-12
> > >
> > > Thank you very much for your detailed feedback and thoughtful review. We greatly appreciate the time and effort you have invested in our paper. Your positive assessment means a lot to us.

---

### Official Review · Reviewer_Ndgx · 2024-07-09

**Soundness:** 2
**Presentation:** 3
**Contribution:** 2
**Rating:** 5
**Confidence:** 4

**Summary:**

The paper presents MOTE, a training-based estimate for Neural Network accuracy, as a proxy method to accelerate Neural Architecture Search. The intuition behind MOTE is the non-convexity and non-linearity in the training loss landscape. Consequently, the authors provide a model that characterizes the training process's non-convexity and convergence speed and consider both in a few-shot joint optimization fashion. The proposed model is applied on a reduced architecture and dataset and requires a few training iterations. The evaluation results are promising and show the superiority of MOTE compared to predictor-based and few-shot NAS methods.

**Strengths:**

- The paper tackles a critical problem—accelerating the NAS framework with few-shot performance predictors.
- The problem is nicely formulated, and the motivations are clearly stated. The paper is also well-written and has a nice flow.
- The evaluation section seems technically sound, with various experiments showing MOTE's superiority compared to state-of-the-art methods.

**Weaknesses:**

- Zero-shot estimates like NTK struggle with high sensitivity to the weights initialization method. While the authors have raised this point, there's no detailed study on MOTE's sensitivity to weight initialization . Furthermore, the type of initialization adopted by MOTE is not mentioned in the paper.
- The non-convexity and non-linearity in DNNs is a core reason for the inefficiency of NTK-based methods. The authors should put more emphasis on this point, as it is a significant factor influencing the performance of MOTE. An ablation study on the impact of skip-connection and non-linear operations on the performance of MOTE can help draw more concrete conclusions about the robustness of the proposed approach when the DNN model is highly non-linear.
- The RD transformation needs to be fully detailed. For CIFAR100, the authors employed a VGG model pretrained on ImageNet1K. However, it's not mentioned whether they perform a fine-tuning for CIFAR100 and the rationale behind the choice of the VGG model specifically or whether VGG should also be employed to perform an RD on any dataset.
- While the comparison in the evaluation section is more focused on predictor-based and few-shot NAS frameworks, it does not specifically compare against zero-shot approaches [1, 2], especially the latest work that considers the feature map correlation [3].
- The search spaces considered are mainly from the NAS-Bench line of work. MOTE should also be evaluated on supernets like P-DARTS and Transformer-based models.
- The MOTE-NAS and MOTE-NAS-EF are built upon evolutionary algorithms. It's therefore unclear whether the superior performance comes from the evolutionary search or MOTE evaluation. The evolutionary search should be compared w/ and w/o MOTE and also against simple baselines (e.g., random search).
- The conclusion section should discuss MOTE's limitations in more details, especially for multi-objective NAS methods, where efficiency and accuracy need to be optimized in a joint fashion.

**References:**
- [1]: Lin, Ming, et al. "Zen-nas: A zero-shot nas for high-performance image recognition." Proceedings of the IEEE/CVF International Conference on Computer Vision. 2021.
- [2]: Bhardwaj, Kartikeya, et al. "ZiCo-BC: A Bias Corrected Zero-Shot NAS for Vision Tasks." Proceedings of the IEEE/CVF International Conference on Computer Vision. 2023.
- [3]: Jiang, Tangyu, Haodi Wang, and Rongfang Bie. "MeCo: zero-shot NAS with one data and single forward pass via minimum eigenvalue of correlation." Advances in Neural Information Processing Systems 36 (2024).

**Questions:**

- To what extent is MOTE sensitive to weight initialization?
- What impact does the non-linearity in DNN architectures have on MOTE’s performances?
- What is the exact process of the DR transformation? Can the same process be adapted to any type of dataset? And what’s the rationale behind choosing a VGG model?
- What if the search space is a supernet (e.g., P-DARTS) or Transformer-based (e.g., NASViT)? How generalizable is MOTE to these search spaces?

**Limitations:**

- Highly non-linear DNNs are more challenging for MOTE---Thus the need for an ablation study on non-linear operations.
- The limited scope of MOTE's evaluation on NAS search spaces—only on NAS-Bench-related search spaces.
- The RD transformation method is highly specific, hindering its applicability to other tasks/datasets.
- No detailed study on the impact of weights initlization.

---

> ### Author Rebuttal · Authors · 2024-08-05
>
> Thank you very much for your valuable feedback.
>
> **Answers to Questions:**
>
> **A1.** To determine whether MOTE is affected by random weight initialization, an additional experiment was conducted, as shown in TABLE II of the attached file. The results indicate that the effect of random weight initialization on MOTE's architecture search is minimal. This is mainly because the landscape and speed terms used in MOTE are derived from a training procedure, making MOTE insensitive to weight initialization.
>
> **A2.** To assess the sensitivity of MOTE to the use of skip connections, we selected four architectures from NASBench-201 to observe changes in accuracy and MOTE values with and without skip connections. These four architectures include Res-Conv(3x3) and Res-Conv(1x1), which use conv-3x3 and conv-1x1 with skip connections, respectively, and NoRes-Conv(3x3) and NoRes-Conv(1x1), which do not include skip connections. As shown in TABLE V and Fig. 2 of the attached file, the speed term, landscape term, and MOTE accurately reflected the changes in accuracy across the four architectures. When the accuracy increases, the values of the two terms and MOTE also increase, and vice versa. These experiments demonstrate that MOTE effectively captures the nonlinear architectures of DNNs.
>
> **A3.** The specific process of RD is discussed in the global rebuttal. We encode each image in the original dataset using a VGG network trained on ImageNet (not fine-tuned on CIFAR-100), inspired by the Inception Score method. Although VGG can be replaced, its inherent knowledge as a simple and high-capacity CNN makes it suitable for image encoding.
>
> **A4.** The MOTE also apply on the open search space(as shown Tab. 2 of our paper), where MOTE-NAS shows significant performance. To evaluate whether MOTE is suitable for other search spaces, an ablation study was performed on NASBench-301 (based on DARTS), as shown in TABLE I of the attached file. In this study, MOTE continues to outperform TSE and Synflow.
>
> **Answers to Weaknesses:**
>
> **AW1.** Please refer to A1.
>
> **AW2.** Please refer to A2.
>
> **AW3.** RD is a method used to accelerate MOTE. The selection method used in RD can be replaced by other methods. The ablation studies, shown in TABLE III of the attached file, indicate that MOTE performs well even if RD is replaced by other methods. Determining the best reduction method will be explored in future research.
>
> **AW4.** In Fig. 7 and Tab. 1 of our paper, most comparison methods such as NASWOT, TE-NAS, KNAS, Zen-Score, and ZiCO are training-free (or zero-shot) NAS methods. However, the term "free" only applies to the search stage; these methods still require GPU time in the evaluation stage. The core idea of MOTE focuses on the time spent in the evaluation stage, which is more time-consuming than the search stage.
>
> **AW5.** Please refer to A4.
>
> **AW6.** MOTE-NAS-RS is a version that replaces the evolutionary algorithm with random sampling (please see Table A4 in the supplementary materials).
>
> **AW7.** The limitations and contributions of this paper have been addressed in the global rebuttal and will be added to the camera-ready version if this paper is accepted.

---

> > ### Comment · Reviewer_Ndgx · 2024-08-12
> >
> > I appreciate the authors' detailed responses, which have addressed several of my concerns. I encourage the authors to incorporate these explanations into the revised version of the paper and to share their codebase as well. I updated my score accordingly.

---

> > > ### Author Response · Authors · 2024-08-12
> > >
> > > Thank you for your kind feedback and for updating your score. We appreciate your suggestions and will make sure to incorporate the explanations into the revised version of our paper. We also plan to publicly share our codebase on GitHub.

---

### Official Review · Reviewer_XpJH · 2024-07-11

**Soundness:** 4
**Presentation:** 3
**Contribution:** 3
**Rating:** 7
**Confidence:** 4

**Summary:**

The paper introduces MOTE-NAS, a novel approach for efficient Neural Architecture Search (NAS). MOTE-NAS suggests a novel proxy utilizing both macro-level loss landscape smoothness and micro-level convergence speed to predict the performance. By utilizing reduced architectures (RA) and datasets (RD), MOTE-NAS achieves state-of-the-art accuracy on benchmarks such as CIFAR-10, CIFAR-100, and ImageNet-16-120 while significantly reducing computational costs.

**Strengths:**

The proxy proposed by the authors has a significant correlation with the final accuracy, showing particularly high correlation in the reduced search space.
The RA and RD methods can drastically reduce the dataset and architecture, providing substantial advantages in terms of speed.

**Weaknesses:**

The design of the skeleton for the reduced architecture seems to require human expert’s guide. It would be better if it were demonstrated that MOTE-NAS is robust across various RA configurations or if there were a general methodology for constructing RA.

**Questions:**

Typos
Table 2, UTRE-NAS

**Limitations:**

The authors do mention some of the limitations in the paper.

---

> ### Author Rebuttal · Authors · 2024-08-05
>
> Many thanks for your high appreciation of our work. We greatly value your feedback.
>
> **Answers to Questions:**
>
> **A1.** If our paper is accepted, all the typos you mentioned will be corrected and eliminated in the camera-ready version.
>
> **Answers to Weaknesses:**
>
> **AW1.** The RA design might be achievable by other algorithms, which remains an open question. However, the proposed RA is simple yet achieves promising results in NABench-101 and NASBench-201, demonstrating its effectiveness across different datasets.

---

> ### Comment · Area_Chair_rRSZ · 2024-08-14
>
> Dear Reviewer XpJH ,
>
> Please respond to authors and engage in discussion as soon as possible.
>
> AC

---

### Official Review · Reviewer_NBd3 · 2024-07-14

**Soundness:** 3
**Presentation:** 3
**Contribution:** 3
**Rating:** 5
**Confidence:** 4

**Summary:**

This paper proposes a novel limited training NAS method that is able to rank the candidate architectures after training them for a limited number of epochs. The MOTE metric consists of two terms, the landscape term and the speed term. The landscape term is indicative of the loss landscape and it is the cross-entropy loss of the model $\theta(g)$ obtained by the weighted sum of the initial parameter weights and the current parameter weights. This loss values are summed for a certain number of epochs. As the number of epochs increase, the weight assigned to the initial model weights decreases and that of the current model increases to result in $\theta(g)$. The speed term is used to estimate the convergence speed of an architecture. It is computed by summing the cross-entropy loss at every epoch divided by the time taken to train for that epoch over all the epochs.

Rather than training the original architectures for N epochs, they derive a reduced variant for each architecture that is close enough to the original architecture. This reduced architecture is then trained on a reduced dataset to obtain the training dynamics necessary for MOTE. The reduced dataset is created by first obtaining the logits of a pretrained VGG-16 model and then for each label, k-means and farthest point sampling are applied to yield the reduced dataset.

**Strengths:**

This paper proposes a novel limited training NAS method that also takes into account the convergence time while training a neural architecture. They were able to consistently achieve better correlation when compared to all the other training-free NAS baselines on 2 search spaces.

**Weaknesses:**

1. What is the correlation between the accuracies of the (i) reduced architectures and the original architectures (ii) training original architectures on reduced dataset and training the original architecture on the entire dataset (iii) the reduced architectures trained on the reduced dataset and the original architectures. Table A3 shows the correlation between the reduced architecture trained for 4 epochs but that is not the actual setting. The correlation already drops from 34.5 to 8.1%. MOTE when applied on the original architectures and the original datasets already would introduce some errors in ranking the architectures. The reduced architecture and the reduced dataset setting would further cascade the errors as the reduced setting is not 100% correlated to the original setting. So how can this reduced setting be used to begin with? Please include the correlation results for all the search spaces and all the datasets.

2.  The authors should not be comparing their approach against zero-cost methods while MOTE considers training information.  Please evaluate the rest of the baselines also on the trained architectures for fair comparison. However, as mentioned above, it is not clear how the other methods would perform in the reduced setting owing to the concerns stated in 1.  Also, 7 seconds per architecture while the rest of the baselines take only 1 second on an average would become very expensive when the number of architectures in the search space is large.

3. Please evaluate the efficacy of MOTE on other search spaces as well such as DARTS, ENAS [1],[2]. [3] demonstrated that training-free NAS methods don't generalize to tasks beyond image classification. Please evaluate on TransNASBench-101 [4] too similar to ZiCO.


[1] NAS-Bench-301and the case for surrogate benchmarks for neural architecture search, Siems et al.

[2] On Network Design Spaces for Visual Recognition, Radosavovic et al.

[3] https://iclr-blog-track.github.io/2022/03/25/zero-cost-proxies/

[4] TransNAS-Bench-101: Improving Transferability and Generalizability of Cross-Task Neural Architecture Search, Duan et al.

**Questions:**

1.  Also, how many architectures did you sample for each search space? Did you run all the other baselines on the same architectures, same data augmentations that MOTE was evaluated on or did you use the results reported in their corresponding papers? Also, if you are evaluating the top-k architectures for some baselines, please change the table to be : (i) evaluation free, top-k=5, top-k=10) for all the baselines and report the accuracies accordingly. Right now the table is inconsistent.
2. How does a linear combination represent the loss landscape? Given a point $\theta_{B}$ in the loss landscape  and 2 other intermediate points $\theta_{C}$ and $\theta_{D}$, if we draw a line from init to B, it might not actually pass through intermediate points in the loss landscape. So how is it capturing the landscape? The main reason this might be working better than TSE is that it is assigning higher weight to the current weights as the epochs increase and lesser weight to the initial weight. How would it compare against TSE-E and TSE-EMA where the loss values of the initial few epochs are discounted.
3. This paper used architecture reduction and dataset reduction to accelerate the training so that MOTE can be computed faster. Can you clarify if the k-means and the farthest point sampling is applied to datapoints corresponding to each label seperately to obtain a subset of them for that label? If not, how exactly is the clustering and the sampling done?
 Can you show how well the architectures the reduced dataset performs when compared to the original dataset by computing their correlations?  Why did you use k-means followed by FPS instead of other subset selection methods such as set-cover, facility location or coresets [5]?
4. Can you please elaborate further what the meta architecture and the reduced architectures are comprised of? It is not clear to me.
5. MOTE uses the landscape term and the speed term. In Table1, the estimation free version of MOTE seems to discover architectures that train much faster than the others. However, once top-k are considered,  the compute cost of training the discovered architecture is not any lesser than those discovered by some of the other baselines. Figure 1 shows that the landscape term performs better than the speed term. Can you also do an ablation and show how the MOTE-landscape and MOTE-speedterm perform in figure 7 and table 1?

[5] https://cords.readthedocs.io/en/latest/strategies/cords.selection_strategies.SL.html#module-cords.selectionstrategies.SL.submodularselectionstrategy

---

> ### Author Rebuttal · Authors · 2024-08-05
>
> Thank you very much for your valuable feedback. However, we noticed that you might have misunderstood the core idea of MOTE. Specifically, the landscape term does not reduce the proportion of $\theta_{init}$ (initial model weights) or increase the proportion of $\theta$ (trained model weights) as epochs increase. MOTE is composed of a macroscopic landscape term and a microscopic speed term. The formula for the landscape term is:
>
> $\theta(g)=(\frac{g}{G})\theta_{init}+(1-\frac{g}{G})\theta,$
>
> where $g$ ranges from 0 to $G$ and is independent of the number of epochs. Here, $G$ is a hyperparameter (default $G=10$), which indicates the granularity of the linear combination between $\theta_{init}$ and $\theta$. Therefore, even if the epochs for training candidate architectures increase, the proportion between $\theta_{init}$ and $\theta$ remains unchanged; only the position of $\theta$ in the loss landscape changes. The landscape term is designed to observe the loss landscape on a macroscopic scale rather than focusing on the minute variations in training loss (as TSE and the speed term do).
>
> **Answers to Questions:**
>
> **A1.** The number of samples ranges from dozens to hundreds. Since we used an evolutionary algorithm for dynamic sampling and conducted each experiment independently ten times to take the average, the specific numbers vary slightly each time. For instance, in NASBench-201, the number of samples for MOTE-NAS-EF ranges from 60 to 100. Some results of our paper in Fig. 1, Fig. 5, Fig. 7 (SynFlow, TSE), and the predictors in Tab. 1 and MobileNetV2/V3 in Tab. 2 are our reproduced experimental results. In contrast, other results directly refer to their corresponding papers. During the MOTE generation process, we did not use any data augmentation techniques (refer to the "GetTrainData" function in the "mote/gen_mote.py" code in the supplementary materials). Not all NAS algorithms follow a "search then evaluate" process; for example, WeakNAS does not follow this process. Therefore, the top-k hyperparameter does not apply to all NAS methods, making it impossible to categorize every NAS method as "evaluation-free" or "top-k".
>
> **A2.** Plotting 2D or 3D loss landscapes with more information is computationally intensive [1], which is not conducive to efficient NAS. Indeed, the linear combination between two points does not fully capture the loss landscape, but it provides a convenient and quick assessment method. This method slices a 1D section through the loss landscape using two sets of model weights, and we can glean some of the landscape features by observing this slice [2]. The increase in epochs does not affect the proportion of $\theta$; please refer to the begin sentence. Our implementation of TSE already includes removing the initial 20% loss values (similar to what TSE-E and TSE-EMA do). In short:
> 1. The landscape term does not increase the proportion of $\theta$ when increasing epochs.
> 2. Since early loss values (20%) are eliminated in TSE, the proportion of $\theta$ increases with the epoch. Therefore, the increase in the proportion of $\theta$ is not why our proposed landscape term performs better than TSE. MOTE aims to describe the loss landscape from a microscopic perspective (speed term) and a macroscopic perspective (landscape term). The speed term can be considered a version of TSE that is aware of time variations.
>
> **A3.** Detailed explanations of RD are addressed in the global rebuttal. The ablation experiments between the original dataset and the reduced one are shown in Fig. 5 of our paper. As the sampling hyperparameter $r$ decreases (RD equals the original dataset if $r=100$), the performance of both the landscape and speed term will degrade, but less than the "accuracy". The core idea of RD is simple and intuitive (see the global rebuttal). Other methods can build RD, the attached file shows the ablation study in TABLE III. Based on the study, MOTE still works well if RD is built by other methods. The best reduction method will be explored in the future.
>
> **A4.** Cell-based search spaces require the combination of the predefined meta-architecture and the candidate cell to form a candidate architecture. The meta-architecture predefines the number of layers, downsampling method, other hyperparameters, etc. Different search spaces have distinct meta-architectures (e.g., NASBench-101 and NASBench-201) with huge time complexities. The goal of RA is to reduce redundant layers as much as possible, except for the cell layers. Although RA results in a simplified structure different from the actual candidate architecture (original meta-architecture + cell), it can construct an extremely lightweight and compact architecture to serve as a new  meta-architecture for NASBench-101 and NASBench-201.
>
> **A5.** MOTE-NAS-EF completely discards the evaluation stage to minimize total time consumption (search time + evaluation time). Since many training-free estimates are unreliable during the search stage, the evaluation stage needs to verify more architectures, leading to higher time consumption (e.g., KNAS). Currently, known training-free estimates are not truly "free" NAS methods.
>
> Other ablation experiments to evaluate the performance of the landscape term and speed term are addressed in the attached file (see Fig. 1 and TABLE IV).
>
> **Answers to Weaknesses:**
> Due to space limitations, these answers will be moved to global rebuttal.
>
> [1]. Li, Hao, et al. "Visualizing the loss landscape of neural nets." Advances in neural information processing systems, 31 (2018).
>
> [2]. Goodfellow, Ian J., Oriol Vinyals, and Andrew M. Saxe. "Qualitatively characterizing neural network optimization problems." International Conference on Learning Representations (2015).

---

> ### Comment · Reviewer_NBd3 · 2024-08-12
> **Response to rebuttal**
>
> I thank the authors for their response. To begin with, while I understand the idea of the MOTE-NAS paper, i seem to have misunderstood the way the linear combination of \theta and \theta_{init} is computed to arrive at \theta_{g}. Thank you for clarifying and correcting my understanding regarding g and G. In that case, how often do you evaluate \theta_{g} during the training?
> While I am familiar with the cell based search space, i now understood what you meant by the term meta-architecture.
>
> A3. I understand that there are various methods to reduce the dataset. I wanted to know how the proposed k-means method compares to other subset selection methods that I pointed out. I see that you compared RD against random sampling in the ablation study.
>
> A5. Algorithms such as TE-NAS, NASWOT, Zico, Zero-cost proxies for lightweight NAS rely only on the inference of a few batches during the search phase and evaluate the best architecture found. Like I mentioned earlier, the architecture found by MOTE-NAS-EF is worse than TE-NAS in table 1. Zico is also not included in Table 1.
>
> While all the search spaces require the algorithm to search for a cell, some search spaces are more complicated than others. NASBench 101, 201 etc are all reduced search spaces when compared to the original DARTS search space. TransNASBench 101 is also a challenging search space. That was the reason, it is important to evaluate how well the proposed method performs on these search spaces
>
> The authors do point out that the correlation of the landscape term and the speed term don't deteriorate as much as the accuracy term in the reduced setting, albeit on 1k architectures. It is important to see how well they can discern between the top few architectures that are close in performance. However, computing the landscape term and the speed term is still computationally expensive.
>
> I thank the authors for answering most of my questions and I would like to increase my score

---

> > ### Author Response · Authors · 2024-08-13
> >
> > We sincerely appreciate the effort you have put into reviewing our paper. We will thoroughly consider your valuable suggestions and carefully reflect on the limitations you mentioned, including computational costs and search space issues. Your insights will significantly contribute to our continued research in the field of NAS. Finally, we would like to express our gratitude once again for your efforts on our paper, and we are also very thankful for your decision to increase the score.

---

### Author Rebuttal · Authors · 2024-08-06

Thank you to all four reviewers for your diligent efforts and valuable suggestions. We appreciate your feedback and comments, which will help us improve the quality of our paper. In this response, we will (1) summarize our paper's contributions and main limitations, (2) address the RD issue raised by the reviewers, (3) respond to the remaining concerns from Reviewer NBd3, and (4) include key experiments to clarify the raised questions.

**Contributions:**

1. Our proposed MOTE-NAS efficiently estimates training outcomes by jointly optimizing landscape view and convergence speed objectives. It captures the non-convex nature of DNNs and monitors convergence speed.
2. We introduce two reduction strategies to accelerate MOTE generation, making the process more lightweight.
3. Our MOTE-NAS establishes a new state-of-the-art accuracy-cost plot for NAS, with an evaluation-free version outperforming some NTK-based methods, such as KNAS.

**Limitations:**

1. Our work currently focuses on image classification; the applicability of MOTE-NAS to other tasks has yet to be explored.
2. While MOTE-NAS has been successfully applied in the closed search spaces (NASBench-101, NASBench-201) and the open search space (MobileNetV3), its effectiveness in broader search spaces, particularly with Transformers, requires further investigation.
---
**To address the reviewers' questions about RD:**

**Q1. Why is RD necessary?**

**A1.** MOTE generation relies on training, which is computationally expensive. For example, 12 epochs on NASBench-201 take about 200 GPU seconds. Reducing the number of training samples can significantly cut this time. RD is designed to accelerate the process by decreasing the number of training samples.

**Q2. Why is it based on CIFAR-100?**

**A2.** Randomly sampling images can lead to underfitting if some labels have too few samples. Sampling by labels instead ensures sufficient sample diversity. CIFAR-100, with many labels and fewer images, is ideal for this method. MNIST or CIFAR-10 would lose diversity at lower sampling rates (e.g., 10%).

**Q3. What is the specific process?**

**A3.** The process involves five steps:

Step 1: Encode each CIFAR-100 image using VGG-16 (trained on ImageNet, without fine-tuning on any datasets), taking the softmax logits.

Step 2: Sum and average the encoding results of images with the same label, resulting in 100 encoding categories for 100 labels.

Step 3: Cluster these 100 encoding categories into $r$ groups using the K-Means algorithm.

Step 4: Extract each center $c_i, 1 \leq i \leq r$ from the $r$ groups.

Step 5: Run FPS (Farthest Point Selection) within each group to find a representative point $l_i, 1 \leq i \leq r$ that is farthest from all $c_i$. Then, {$l_i$} is the set of representative labels required by RD.

Steps 3 to 5 aim to avoid selecting similar labels (e.g., bus and streetcar). We first group labels with K-Means and then use FPS to select labels that are farthest from all group centers.

**Q4. Is RD effective? Are there other methods?**

**A4.** RD has shown high compatibility with MOTE, significantly speeding up calculations despite some loss in performance (refer to Fig. 5). An intuitive method (RD-RS) that randomly samples images was also tested and is shown in the attached PDF file. As shown in TABLE III, although RD outperformed RD-RS, the latter still performed well, indicating that alternative methods for generating RD are possible and inspire further research.

---
**Weaknesses mentioned by Reviewer NBd3:**

**AW1.** Both reduction strategies (RA and RD) accelerate computation but harm performance, affecting both metrics 'accuracy' and 'MOTE'. However, MOTE's performance drops less than 'accuracy' (see Tab. A3). Despite the errors caused by the reduction strategies using RA and RD, MOTE can maintain a high correlation and significant acceleration. The insight of this paper is to prove MOTE's ability to capture the loss landscape from macro and micro perspectives, using the reduction strategy for computation speed gains. Experiments (Fig. 5 and Tab. A3) show that 'Accuracy' adapts poorly to this strategy, while MOTE's performance and speed impact are good trade-offs. Further experiments (Fig. 7, Tab. 1, Tab. 2) illustrate MOTE results with RA+RD applied, showing its effectiveness across different benchmarks, including NASBench-101, NASBench-201, and MobileNetV3 search spaces.

**AW2.** Not all comparison targets are training-free; methods such as TSE, LGA require training. The primary comparison, NTK-based estimates, though training-free, are unstable, as shown in Fig. 1, and a similar conclusion is supported by the LGA paper. The reduction strategies will degrade performance, so current comparison targets without reduction strategies represent their upper-performance bounds. Under this adverse situation, MOTE achieves a higher correlation (Fig. 7), making further reduction strategies and applications on other estimates unnecessary. On the other hand, evaluating the final performance of a searched architecture is very time-consuming (e.g., full training on ImageNet requires days to tens of days of GPU time). While MOTE takes 7 seconds, it is more effective than other estimates in the search stage for finding promising architectures, reducing the burden of the evaluation stage. Consequently, the overall time consumption is faster than other training-free methods (as shown in Tab. 1 and Tab. 2).

**AW3.** Our experiments in Tab. 2 were performed based on an open MobileNetV3 search space, employing a variation of RA—Rescale Reduced Architecture—for searching promising architectures (see section A.6 and Fig. A4). Additional experiments on NASBench-301 (based on DARTS) are included in the attached  file (please refer to TABLE I). Most NAS tasks focus on searching the backbone for image classification. Expanding NAS tasks is crucial in NAS research and not easily finished during the rebuttal time. We plan to explore this further.

---

### Decision · Program_Chairs · 2024-09-25

**Decision:**

Accept (poster)

**Comment:**

This paper proposes to use both macro-level loss landscape smoothness and micro-level convergence speed as proxy for efficient neural architecture search, improving the accuracy and cost trade-off. The proposed method is effective in building a good correlation between the proxy and the final accuracy. Overall, the paper is well-written. Most reviewers are satisfied with the authors’ response to their concerns. Therefore, the AC recommends Accept. The authors should incorporate the comments and suggestions from reviewers when preparing the final version of the paper.